

# Silica diagenesis-driven fracturing in limestone: an example from the Ordovician of Central Pennsylvania

Emily M. Hoyt[1], John N. Hooker[1]

[1]The Pennsylvania State University, Department of Geosciences, 503 Deike Building, University Park, PA 16802, USA

*Correspondence to*: John N. Hooker (jzh497@psu.edu)

**Abstract.** Fracture patterns, interactions, and crosscutting relationships are tools for interpretation of fractures as paleostress indicators for past tectonic events and as past or present-day fluid-flow networks. In the Appalachian Basin in Central Pennsylvania along Mount Nittany Expressway Route 322 lies a significantly stratified fracture set hosted in Ordovician age limestone. Tectonic strain is a problematic mechanism for these fractures because they are hosted in individual beds lacking

apparent mechanical significance relative to other limestone beds in the outcrop. Many of the fractures are layer-parallel, a characteristic commonly observed in shales, due to shales' mechanical anisotropy and tendency to develop fluid overpressures; however, these fracture-hosting limestones lack obvious mechanical anisotropy. Fracture orientations vary, but desiccation, bentonite swelling, and dolomitization are eliminated by an interpreted transgressional paleoenvironment and a deficiency of the hypothesized minerals.

X-ray diffraction determined the composition of samples collected, point-count quantification determined fracture intensity, and optical petrography recorded scaled petrographic photographs. Comparison between fracture intensity and host-rock minerals reveal that silica content is consistently depleted in fractured layers relative to unfractured layers. The diagenetic transition of biogenic silica to quartz is suggested to be the driving mechanism based on silica being present as biogenic grains, as well as cement and detrital grains, and fractures being filled with calcite cement. Silica migration explains the

volume lost from fractured layers in a proposed horizontal fracturing mechanism whereby the host rock shrinks but is excluded from vertical contraction.

## 1 Introduction

Rock fractures are significant geologic features for the fields of hydrology, engineering, mining, and energy. Fracture interactions, patterns, and crosscutting relationships allow interpretation of fractures as paleostress indicators for past

tectonic events (Price & Cosgrove, 1990). As fractures form and evolve, they can drastically alter the host rock's permeability as the fracture creates a void region in the host rock (Wolfsberg, 1997). Fractures generated by induced mechanical stress, are predominantly recognized in the literature; however, chemically driven fracturing is underappreciated. Chemically generated fractures have been documented, though principally in limestones through the generation of carbonate minerals by dolomitization (Bellamy, 1977). Chemically induced alterations to rock permeability are significant for carbon

sequestration and seal capacity (Min et al., 2009).





In the present study, we focus on a significantly stratified fracture set hosted within limestone in the Appalachian Basin in Central Pennsylvania along Mount Nittany Expressway Route 322, (Fig. 1, Fig. 2). These beds lack obvious mechanical significance relative to the other limestone beds in the outcrop. A tectonic origin is problematic as folds and faults are spatially associated with additional fractures throughout the outcrop, yet those fractures are not significantly stratified.

Chemical reactions associated with volume changes, and therefore capable of driving rock fracture, include dolomitization, silica diagenesis, evaporite precipitation, dehydration, and bentonite swelling. To test whether any of those reactions might have produced layer-parallel, stratified fractures here, we looked for the correlations between fracture intensity and host-rock minerals using field measurements of the fractures and XRD analyses of host rocks.

**Geologic Setting**

The overall sequence represents the evolution from a passive to an active, convergent margin during the late Ordovician. Numerous cycles are observed throughout the beds beginning with passive-margin carbonate platform sediments which progressively intertwine with siliclastic influx from an uplifting source, representing the transition into a foreland basin. Limestone beds include calcareous tempestites which are products of long-term storm events whereas the argillaceous limestone and shale interbeddings are an index of a slow background sedimentation in a hemipelagic setting. This indicates a

deep to mid-ramp parasequence in which bentonites accumulated on some flooding surfaces; possibly a mechanism of eustatic sea level cycles (Gold et al., 2017).

The collision of microcontinental and Theic assemblages with Laurentias eastern margin ultimately generated the Taconic orogeny during the Late Ordovician. This orogeny influenced the initiation of the Appalachian orogen as the tectonic foundation of Laurentias east margin, and the Early Paleozoic carbonate shelf structure was significantly transformed (Faill,

1997). The southeast region of the Valley and Ridge fold belt and the northwest Appalachian Plateau fold belt region are divided by the Allegheny structural front. A slip event produced a fault-bend fold structure, known as the Nittany Anticline and Bald Eagle Mountain, dipping to the south and establishing major detachment horizons. The structural front in north-central Pennsylvania is underlain by a southeast dipping thrust ramp (Mount, 2014).

Here we focus on highly stratified, layer-parallel fractures hosted in limestone beds within the transition from the upper

Salona Formation to the lower Coburn Formation (Fig. 1, Fig. 2). Layer-parallel fractures in the nearby downsection Bellefonte and Loysburg Formations limestone were attributed to Alleghenian-age deformation (Srivastava & Engelder, 1990). The present layer-parallel fractures do not contain fluid inclusions sufficiently large for paleothermometry. Moreover, the fracture cements do contain significant twinning, so any temperatures gleaned from such cements would have dubious value, because of potential deformation of inclusions, post-sealing.



## Methods

### X-Ray Diffraction

X-ray diffraction determined the composition of the samples of fractured layers, the unfractured limestone and clay layers that contact the fractured layers, and unfractured layers away from fractured layers (*table 1*).

The X-ray diffractometer used in this study was a multi-purpose diffractometer instrument, the Malvern Panalytical X'Pert Pro MPD theta-theta Diffractometer. The program used to analyze the results of the XRD scan was MDI Jade that evaluates XRD patterns and interprets them for weight percentages and phase classification. To assist in identification of clay minerals, one sample (CA6-decarb) was powdered and rinsed in dilute acetic acid to remove carbonate. There is an error of +/- 5% of each reported value placed on XRD data.

### Point Quantification

Point quantification determined fracture intensity of the fractures hosted in each bed. Square grids were printed over field photographs, and the volumetric fraction of a host rock comprising its intruding veins was calculated as the number of grid intersections that fall on veins divided by the total number of rock intersections within the grid. This value is here termed the fracosity.

Optical petrography was performed using a polarizing microscope, the Zeiss Axio lmager A2.m, with ×2.5, ×10, ×20 and 75 ×50 objectives and an AxioCam 105 color camera system. The program Zen was used to record scaled petrographic photographs.

## Results

### Outcrop description

Limestone layers, including fractured layers, stand prominently in relief while argillaceous limestone, clay, and shale 80 weather recessively. The data for each type of strata is represented as a percentage in a pie chart of the summed bed thickness throughout the interval within each of six sub-intervals (A through F, *Fig. 2*). There is no obvious relationship between the abundance of fractured layers and the accompanying facies. The fractured layers are present at roughly regular intervals in the outcrop despite varying proportions of limestone and shale in the section.

Strike and dip measurements were recorded of bedding, layer perpendicular factures, and layer parallel fractures. Layer-85 perpendicular fractures (*Fig. 3*) are present throughout the limestone layers and absent within shale layers. Such fractures are more common farther downsection and are not particularly abundant within fractured layers one through six.

Numerous fractures in the outcrop are associated with tectonic faults and are commonly preferentially hosted within limestone layers but not otherwise stratified (*Fig. 4*). These fractures have various orientations, including ones sub-parallel to



large faults and having significant shear displacement. As well, these fractures may merge or bifurcate to form closely
spaced networks having a cataclastic texture (*Fig. 4*).

A subset of the observed fractures is found to have a highly stratified pattern and, unlike the other fracture measurements, a layer-parallel orientation. Six "fractured layers," F1 through F6, moving downsection, contain abundant layer-parallel fractures and subsidiary layer-perpendicular fractures. F1 through F6 lie within the transition between the Salona and Coburn Formations, whose contact is uncertain and probably gradational within this roadcut. These fractured layers are observed in
the southeastern limb of the Mount Nittany Syncline but cannot be directly correlated due to safety concerns within the quarry. Intervening layers, including limestone layers, show only sparse fractures, of any orientation.

In fractured layers 5 and 6, there is a stratiform color change within the limestone beds. The fractured beds of fractured layer 5 (*Fig. 5*) are observed as a medium grey limestone that is underlain by a sharp shift to a tan limestone with black and brown clay clasts. The top $\frac{2}{3}$ of fractured layer 5 are gray and fractured, whilst the bottom $\frac{1}{3}$ is pale gray with banding.
However, in both fractured layers 5 and 6, the tan limestone then grades downward to a clay layer.

**Fracture Orientation**

The bedding of the outcrop and the poles to the fractures are shown in a stereonet *(Fig. 6)*. Bedding dips gently (15-20°) to the east. Fractures are dominantly layer-parallel, but layer-perpendicular fracturs are common. Within F1 though F6, layer-parallel fractures may be present in the absence of layer-perpendicular fractures, but the latter are not encountered without
the former. There is no apparent dominant strike orientation for bedding-perpendicular fractures. There is a potential correspondence of these fractures with the J1 and J2 set of Pennsylvania's regional structural geology (Engelder, 2004), however, additional measurements are needed to rigorously determine an association. Layer-parallel and -perpendicular fractures show mutual crosscutting within F1 through F6.

**Fracosity results**

Results of point counts, taken to quantify fracture volume (fracosity), are shown in *table 2*. The total area represents the total points within the field picture; the N/A area being the points of neither a fractured or unfractured point of the photo such as dirt, moss, or a shade; the unfractured area represents the unfractured, or host rock, points in the photo. Fracosity ranges from 1.29 to 6.15% within fractured layers with an average of 3.36%. The minimum fracture volume hosted within a fractured layer, 1.29%, is used as a conservative upper limit to for the fracture volume that might be hosted in the
unfractured layers, but undetected.

**XRD results**

XRD results indicate that the silica content is consistently depleted in fractured layers F1 through F6 in comparison to the nearby unfractured layers. However, the unfractured tops and bases of the fractured layers have the highest quartz





composition. Albite does not systematically vary throughout each group, but albite content varies with a range of 0-3.9 wt. %

(*table 3*). The albite composition from each layer is not observed to correlate with quartz (*Fig. 7*) such that the quartz to albite ratio is the lowest in the fractured layers (additional mineral compositions in *Appendix 1*).

The mineral compositions (in wt.%) of quartz and calcite, from the individual fractured and unfractured layers, as measured using XRD, are plotted against fracture volumes in *Fig. 8*. There is a negative correlation between fracosity and quartz content, and a positive correlation between fracosity and calcite content.

The individual fractured layer thickness displays no obvious correlation with the fracosity percentage for the corresponding layer.

These observations are consistent with field photographs of fractured layers. A field photo for fractured layer 2 is shown in *Fig. 9* and for fractured layer 6 in *Fig. 10*. These photos display the layer-parallel and perpendicular attitude of the fractures, and their mutual crosscutting. The representative fracture volume can be observed in the photos in which layer 2 has

significantly more fracosity, mostly manifest in greater apertures, in comparison to layer 6. These observations are quantified in *table 2* which displays the fracosity for each fractured layer.

**Petrography**

Petrography revealed quartz present in at least three forms: detrital, pore filling, and skeletal grains. The thin section observed in *Fig. 11* originates from a sample collected from fractured layer 1. Quartz in *Fig. 11* appears as a clear mineral

lacking cleavage and having gray or white interference color under crossed polars. Here, quartz has a euhedral termination with abundant solid and fluid inclusions, and is surrounded with a carbonate mineral, based on the limestone lithology and the mineral's higher-order birefringence.

In the silicified base layer underneath fractured layer 2 there is a presently silica test *(Fig. 12)* and in fractured layer 1 there is a dissolved test *(Fig. 13)*; in the silicified base layers, silica is also present as disseminated pore-filling cement. These

observations are consistent with the fractured layers being sites previously occupied by silica that has since migrated from these layers and left behind voids which have been replaced by calcite. The result of this migration can be observed in *Fig. 9* in which fractured layer 2, hosting 5% quartz, is underlain by a silicified base layer, hosting 15% quartz.

The bulk mineral composition of the base of layer 2 is richer in quartz compared to fractured layer 2, which lies immediately above, and to fractured layer 1. The boundary in thin section for the base of layer 2 and fractured layer 2 can be observed in

*Fig. 14*. On the left side of the photo, a dark band of abundant silica grains are observed to abruptly transition to a paler layer lacking abundant silica. This change in mineralogy within the fractured layer 2 coincides with a change in color apparent in the outcrop (*Fig. 9*). The fractured central part of the layer is light gray in color whereas the unfractured base of the layer has a tan, brown color. The same pattern is reflected in fractured layer 6 *(Fig. 10)*, in which the bulk mineral composition of the base of layer 6 is richer in quartz compared to fractured layer 6 (*table 3*).

A petrographic photo of fractured layer 1 with a crosscutting fracture is observed in *Fig. 15*. The fill material is sub-angular and blocky, indication void-filling cement. The crosscutting relationship between the largest two fractures is ambiguous,



with a series of cement blocks within the large, horizontal fracture apparently connecting the separated vertical fracture. It may be that the two fractures were open at the same time; the horizontal fracture appears to crosscut smaller vertical fractures. The presence of microfractures *(Fig. 14, Fig. 15)* suggests that our field-photograph based fracosity Fig.s should

be regarded as minimum estimates.

**Discussion**

The layer-perpendicular fractures in this study—apart from those that from as a subsidiary set alongside the layer-parallel fractures—preferentially form within brittle, stiffer layers. Commonly, as layer-parallel extension increases sequential infilling occurs as a result which induces new fractures to form between existing fractures. Fracture spacing is inversely

related to the applied layer-parallel strain and expected to diminish as the overburden pressure increases and the tensile layer strength decreases (Schöpfer et al., 2011). Such an explanation is consistent with most layer-perpendicular fractures in this study. However, the layer-extension model is unsatisfactory for the arrays of stratified fractures present in fractured layers one through six, because (i) those fractures are dominantly layer-parallel, and (ii) there is no obvious stiffness contrast between fractured and unfractured limestones, judged by the abundance of other fracture sets in the outcrop, which readily

form in any limestones and are only absent from the intervening shales.

The first proposed fracture mechanism is the effect of stress and strain in the rock layers experienced during tectonic events, such as folding in the Alleghenian orogeny (Faill, 1997). However, a tectonic interpretation suffers from numerous inconsistencies for the fractured layers because numerous fractures in the outcrop are associated with tectonic faults and are not stratified *(Fig. 4)* or are present throughout the limestone layers and absent within the shale layers *(Fig. 3)*. This contrasts

with catagenesis (Flores, 2014) which could induce volume swelling and produce fracture; though, such fractures would be expected to preferentially form in organic matter preserving shale layers (Ma et al., 2017). The theory of systematic joints in sedimentary rocks predicts that joint abundance is proportional to the inverse of the bed thickness (Hobbs, 1967). However, the fractures are not strongly aligned in any systematic direction with respect to the fold *(Fig. 6)*. The final mechanism considered is diagenesis, because the problems associated with physical explanations point us toward chemical principles.

Based on these fractures' bed-parallel attitude, we look for a source of internal pressuring, which would explain why they apparently open against gravity (Cosgrove, 1995). An increase in fluid pressure could be generated by disequilibrium compaction, hydrocarbon generation, or a combination of the two (Cobbold et al., 2013). Disequilibrium compaction arises where porosities are higher than expected at a given depth and deviate from the standard porosity trend (Zhang, 2013). The process of hydrocarbon generation increases fluid pressure in low permeability shales as kerogen is buried and heated (Ma et

al., 2017). In either case, high pore fluid pressure produces overpressure, a state where the fluid pressure is greater than the hydrostatic pressure at a given depth. If the fluid pressure exceeds the overburden stress then horizontal fractures can form (Cobbold et al., 2013). The fractured beds observed in this study are hosted in limestone that lack significant organic





material or signs of disequilibrium compaction, such as soft-sediment deformation. In the absence of likely candidates for fluid overpressure, it may be that a geochemical reaction forced the fractures to open.

The diagenetic growth of dolomite can also be a primary expansional mechanism (Bellamy, 1977). However, dolomitic beds are a minor component of the studied formation. Tension fractures are known to form as a result of shrinkage within the rock due to contraction of mud or silt sediments through desiccation (Singhal & Gupta, 2010). However, the fractures in this study do not host mudcrack geometries, nor does the depositional environment display evidence of experiencing extreme dryness. The XRD results detected no traces of evaporites and the paleoenvironment is interpreted to be a transgressional

phase. Fractures can form as bentonite absorbs water into the interlayer region of the crystal lattice. However, the style of fracturing present in the fractured layers is not observed near documented bentonite beds (Gold et al., 2017).

Throughout the stratigraphic column, there is varying silica content, and a correlation is observed in which the fractured layers contain low silica content and a high calcite content. The XRD results demonstrate a significant quantity of quartz hosted in each layer, excluding the acid washed clay layer. These results also yield that the silica is consistently depleted in

all the fractured layers throughout the outcrop when compared to nearby unfractured layers. This produces a negative correlation in which fracosity increases as silica content decreases *(Fig. 8)*.

The quartz could be detrital; however, most quartz observed through petrography is in the form of cements or biosiliceous tests that have become mobilized silica during the transformation to quartz at depth *(Fig. 12, Fig. 13)*. This observation is consistent with the primary source of silica being biogenic and not detrital. A significant quantity of detrital quartz is present,

based on petrographic analysis *(Fig. 11)*. We assume that essentially all albite grains are detrital and so reflect siliciclastic deposition, which would have been accompanied by detrital quartz. This conjecture is consistent with albite composition plotted versus quartz composition *(Fig. 7)*, in which there is no overall correlation but a positive correlation between quartz content and albite content within the sub-populations of both the unfractured and fractured layers.

An explanation for the stratified fracture pattern should account for the systematic paucity of silica within those layers (i.e.,

the "de-silicified" fractured layers) and the abundant silica within adjacent beds. Many possibilities exist, but we suggest that silica diagenesis drove fracturing by either (i) modifying the mechanical properties of the host layers, such that they became susceptible to fracturing by imposed tractions, (ii) modifying the hydraulic properties of the host layers, such that they became susceptible to fluid overpressures, or (iii) producing volume changes within the host layers, directly causing stresses and driving fractures open as a result.

A significant present-day mechanical property contrast is unlikely, for the reason discussed above: fractures are abundant elsewhere in the section which are confined to brittle limestone layers, and absent from intervening shales. However, we can only guess at the mechanical properties the beds would have had during the transition from biogenic silica to quartz, which occurs during burial to relatively shallow depths, at temperatures less than 80°C (Davies & Cartwright, 2002). Precipitation of quartz would likely have coincided with mechanical consolidation if the sediment were not consolidated by that time.

Lithification of a pair of relatively silicified limestone beds, and their bonding to an unsilicified limestone bed in between, under confining stress, might have produced residual stresses that led to fracturing once the stress state changed (Bourne,



2003). Likewise, silicification may have reduced permeability to create local overpressures, assisting the development of disequilibrium compaction. However, both of these processes would seem equally capable of forming fractures in unsilicified units underneath the lowest silicified bed, and yet such fractures are not observed.

Silica diagenesis has been proposed to create opening-mode fractures (Hooker et al., 2017). The correlation between the fractures and silica content suggests that diagenesis could have generated fracturing the dissolution and reprecipitation of biogenic silica. Mobilized silica would produce quartz in the form of lightly silicified layers and result in a fracturing at stratigraphic levels that are dominated by dissolution and thus, volume loss. Hooker et al (2017) presented evidence that silica diagenesis can explain fracture volumes of several percent; we find similar results here (see calculations in *Appendix*

*2*). However, fractures in that study were almost exclusively layer perpendicular. The explanation given was that volume loss within a laterally pinned rock layer resulted in subvertical failure planes. A similar process could have produced the present fractures, but their layer-parallel orientation implies that the host rock would have been vertically pinned—that is, prevented from contracting vertically.

Energetically, the meaning of "vertically pinned" is that the energy required to collapse the de-silicified layer, $S_C$, is greater

than the energy required to generate fractures ($2G$, where $G$ is the surface energy of the rock), plus any strain energy required in association with fracturing, $S_F$. $S_C$ and $S_F$ may be manifest in the form of elastic deformation, grain-boundary sliding, shear microfracturing, mineral solution, or crystal plasticity. We assume that $S_F$ arises from internal deformation of the layer, which does not shorten vertically. As such, collapse of the de-silicified layer would entail a relative energetic benefit, $H$, by lowering the overlying rock mass by the distance to which the de-silicified layer is shortened. We assume that the fractures

formed in lieu of this lowering, and so can treat this lowering distance as equivalent to $h$, the cumulative aperture of horizontal fractures:

$$H = \rho g z \times h \tag{1}$$

where $\rho$ and $z$ are the density and height of the overlying rock, respectively, and $g$ is the acceleration of gravity.

For horizontal fracture development to be energetically favorable over the collapse of the de-silicified bed, we therefore should meet the criterion:

$$2G + S_F < S_C - H \tag{2}$$

The units of each term are energy per unit area; the two strain terms ($S$) are volumetric strain energy times layer thickness. Re-arranging (2), we can state that the energetic cost of shape change in the form of collapse, in excess of that for shape change in response to fracturing, is, at least, the sum of the fracture surface energy and the potential-energy benefit of

lowering the overburden by collapse:





$$S_C - S_F > 2G + H \tag{3}$$

The right side of (3) can be estimated using $G \sim 10$ J/m$^2$ (Friedman et al., 1972), $\rho < 2700$ kg/m$^3$, $h < 10^{-2}$ m in most beds
(*Fig. 9*, *Fig. 10*), and $z < 3000$ m for the depth of the silica transition (Davies and Cartwright, 2002). A high-end estimate of the right side of (3) is therefore on the order of $10^6$ J/m$^2$, with $G$ being negligible compared to $H$. We have little constraint as to the left side of (3), but we can estimate the range of elastic strain energy that would be required to vertically contract the host material, and thus give an estimate of $S_C$ assuming elastic behavior. Young's moduli for clays and soils range from approximately $1 \times 10^6$ to $1 \times 10^7$ Pa, and limestone ranges from approximately $2 \times 10^{10}$ to $6 \times 10^{10}$ Pa (Gudmundsson,
2011). To achieve a 1% shortening strain (*Fig. 9*, *Fig. 10*) would require between $1 \times 10^4$ and $6 \times 10^8$ Pa of pressure. Layer thicknesses on the order of 0.1 m imply an energy requirement between $1 \times 10^3$ and $6 \times 10^7$ J/m$^2$. Therefore, the high-end estimate of the right side of (3) is considerably lower than the high-end estimate of the left side. So, if there is a large difference between $S_C$ and $S_F$, the fracture mechanism of volume loss amid vertical pinning is plausible.

**Conclusion**

Outcrop, petrographic, and mineralogical observations in Ordovician limestones in Pennsylvania show that stratified fractures are located in beds that lost silica, likely during the diagenetic transition of biogenic silica to quartz. Silica is observed as dominantly biogenic grains and cement, with subsidiary detrital grains, and there is an anti-correlation between fracture volume and silica content. Fractures are observed in many orientations and are poorly explained by formation in response to tectonic strains or fluid overpressures. A transgressional environment rules out a desiccation mechanism. The
fractures are not related to bentonite deposition, and dolomitization is ruled out via the absence of dolomite in XRD. Migration of silica explains the volume lost from the fractured layers; calcite cement currently fills the fractures. The details of how volume loss produced layer-parallel fractures are unclear, but a mechanical analysis indicates such a mechanism is possible if the host rock is prohibited from vertical contraction during volume loss.

**Data availability**

Data from this study are listed in Appendix 1.

**Author contribution**

EMH performed field work, point counts, and XRD and petrographic analysis, and wrote the paper. JNH designed the experiment and contributed to field work and manuscript editing.



**Competing interests**

The authors declare that they have no conflict of interest.

**Acknowledgements**

Funding was provided by the Penn State Department of Geosciences. JNH is funded by a fellowship of the GDL Foundation.
We are grateful to Duff Gold, Charlie Miller, Terry Engelder, Liz Hajek, and Mark Patzkowsky for helpful discussion, to
Keith Bowes and Dave Lomison for quarry access, to Nichole Wonderling, Beth Last, and Gino Tambourine for XRD
assistance, and to Andy Smye for microscope assistance and discussion.

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






**Figure 1: The area of study with accompanying stratigraphic nomenclature and descriptions (modified from Berg et al., 1980).**







**Figure 2: Stratigraphy column of the outcrop with according rock classification pie chart. The pie chart labels are as follows: L limestone, S shale, AGL argillaceous limestone, C clay, and F fractured layer. Triangles show the locations of fractured layers; circles show the locations of sampled unfractured layers. Pie chart components are weighted by total stratigraphic thickness within each sub-unit (A through F).**

**Table 1: The nomenclature and description of the fractured, unfractured, and clay layers. Locations of samples shown in Fig. 2.**

| Layer | Notes |
|---|---|
| F1 | Fractured layer 1, grey limestone, muddy, no lamination, continuous styolites, box work fractures, layer parallel fractures |
| U1.5 | Unfractured layer between fractured layers 1 and 2 |
| 2-base | Unfractured dark gray layer below fracture layer 2, thin section of contact |
| F2 | Fractured layer 2 |
| U2.5 | Dark gray massive tops & pale gray laminated bases w/ burrows in 2nd layer, unfractured layer between fractured layer 2 and 3 |
| CA2 | Clay above fractured layer 2 |
| F3 | Fractured layer 3 |
| U3.5 | Unfractured layer between fractured layers 3 and 4 |
| F4 | Fractured layer 4, lightly fractured, right above recessive band below which banding of outcrop |
| U4.5 | Unfractured layer between layers 4 and 5 |
| F5 | Fractured layer 5, top 2/3 gray and fractured, bottom 1/3 pale gray and banded, low fracture density |
| U5.5 | Unfractured layer between layers 5 and 6 |
| CA6 | Clay above F6 in between F6 & UF5.5 |
| CA6-decarb | Clay from CA6, crushed, washed in acid to remove carbonate |
| 6-base | Unfractured dark gray layer below fractured layer 6, thin section for contact 6-base & F6 |
| F6 | Fractured layer 6, medium gray limestone then sharp shift |
| 6-top | Layer above UF5.5, tan Layer above F6, 6Top: gray regular limestone above that |



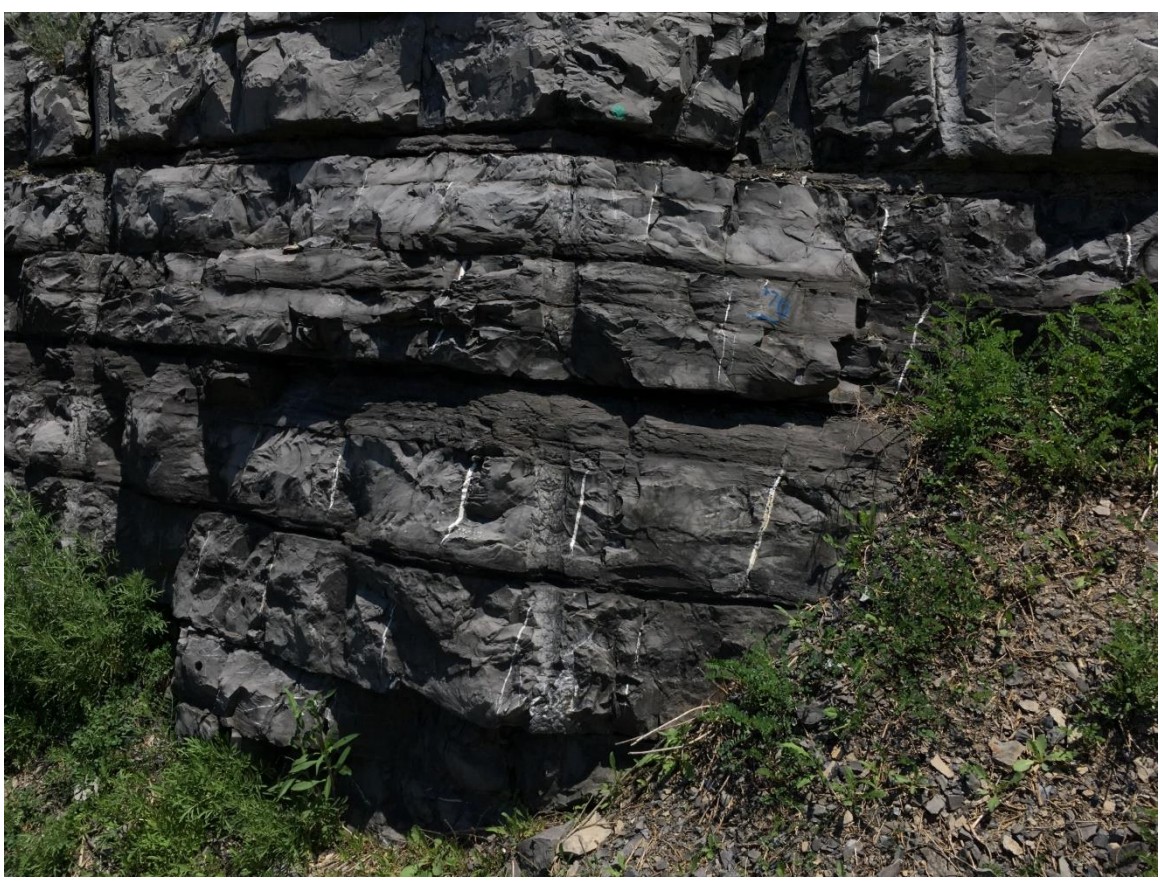

**Figure 3: Regular vertical fractures observed in limestones and absent in shale layers.**



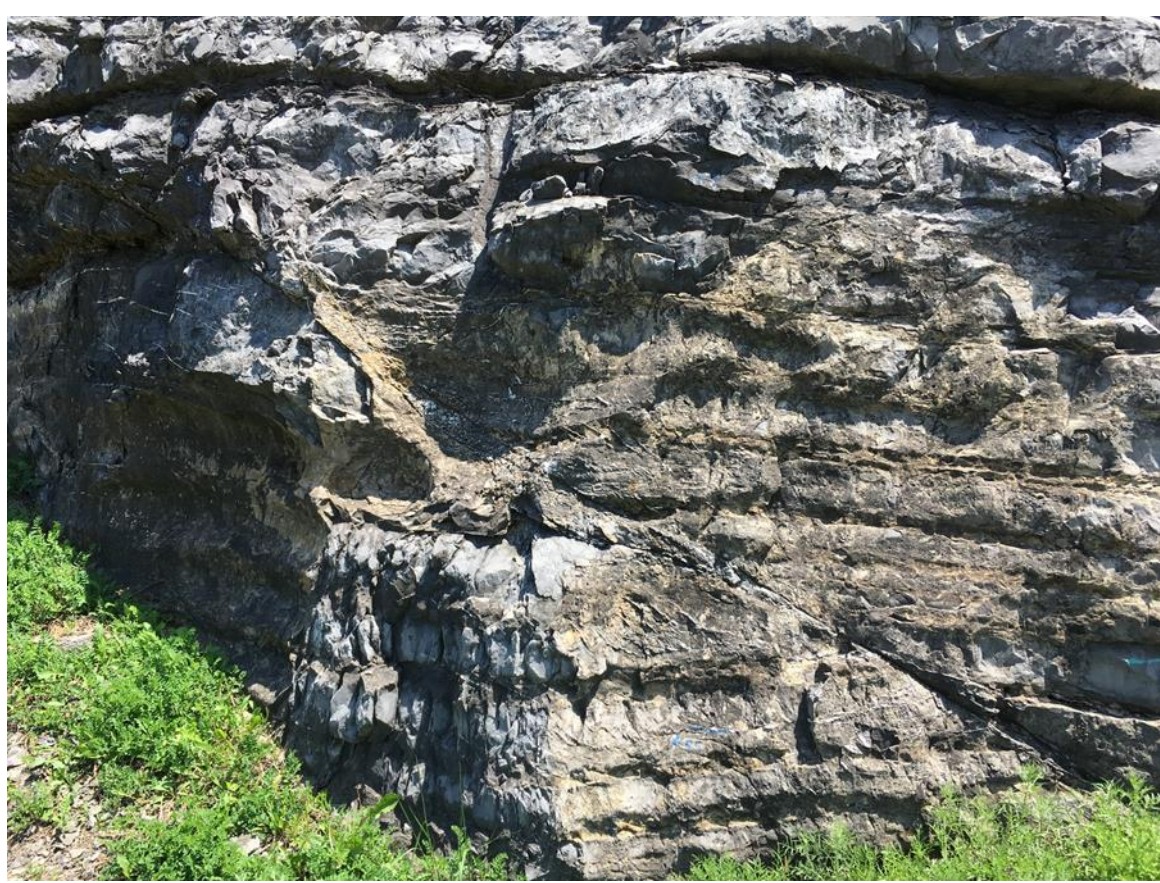

**Figure 4: One of the observed major tectonic faults located in the outcrop. There are abundant opening-mode fractures in its vicinity.**

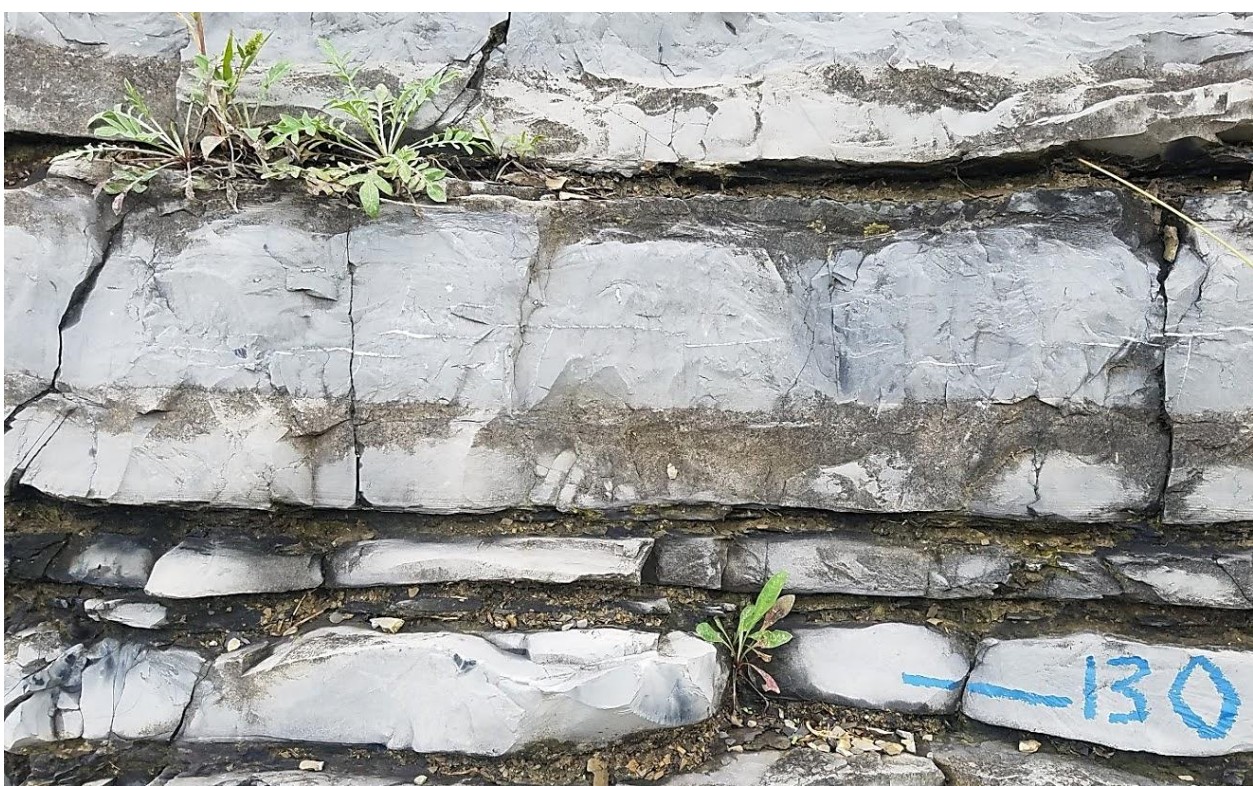




**Figure 5: Field photo of fractured layer 5, hosting predominantly layer-parallel fractures, with unfractured layers located above and below.**

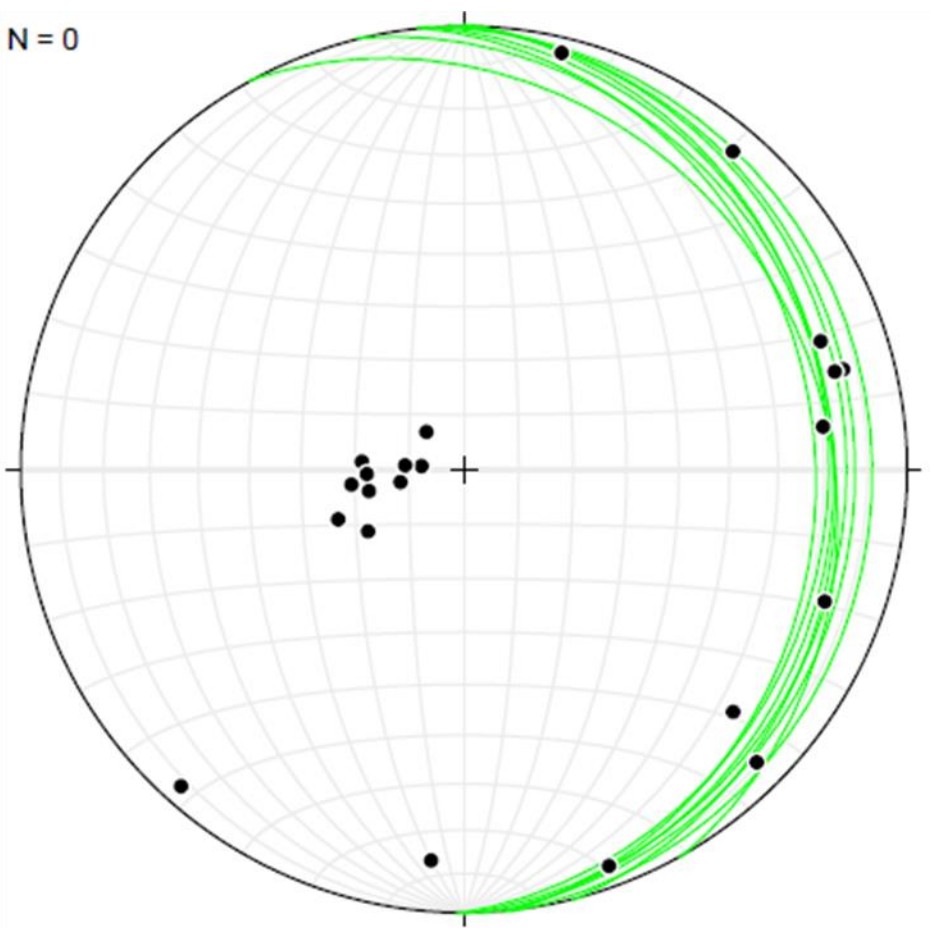


**Figure 6: Stereonet with bedding as green planes with poles to fractures as black dots.**

**Table 2: Results of point counts for each fractured layer and the calculated fracture volume as fracosity.**

| Fracture Layer | Total Area | N/A Area | Fractured Area | Unfractured Area | Fracosity % |
|---|---|---|---|---|---|
| 1 | 7,367 | 424 | 257 | 6,686 | 3.49 |
| 2 | 10,584 | 182 | 651 | 9,751 | 6.15 |
| 3 | 5,472 | 351 | 300 | 4,821 | 5.48 |
| 4 | 3,655 | 268 | 47 | 3,340 | 1.29 |
| 5 | 12,024 | 134 | 226 | 11,664 | 1.88 |
| 6 | 8,500 | 133 | 157 | 8,210 | 1.85 |






**Table 3: Quartz and albite composition (wt.%) and quartz to albite ratio for corresponding layer.**

| Layer | Quartz | Albite | Quartz:albite | Fracosity % |
|---|---|---|---|---|
| F1 | 3.9 | 1.8 | 2:1 | 3.49 |
| U1.5 | 10.4 | 1.3 | 8:1 | |
| 2-base | 15 | 0.7 | 21:1 | |
| F2 | 5 | 1.1 | 5:1 | 6.15 |
| U2.5 | 13 | 1.4 | 9:1 | |
| CA2 | 11.4 | 0 | | |
| F3 | 7.3 | 2.3 | 3:1 | 5.48 |
| U3.5 | 14.7 | 1.9 | 8:1 | |
| F4 | 9.5 | 3.9 | 2:1 | 1.29 |
| U4.5 | 11.6 | 0.7 | 17:1 | |
| F5 | 8.2 | 2.1 | 4:1 | 1.88 |
| U5.5 | 14.6 | 1.6 | 9:1 | |
| CA6 | 17.8 | 1.7 | 10:1 | |
| CA6-decarb | 0 | 0 | | |
| 6-base | 17.1 | 1.3 | 13:1 | |
| F6 | 7.9 | 1.7 | 5:1 | 1.85 |
| 6-top | 14.4 | 1.2 | 12:1 | |

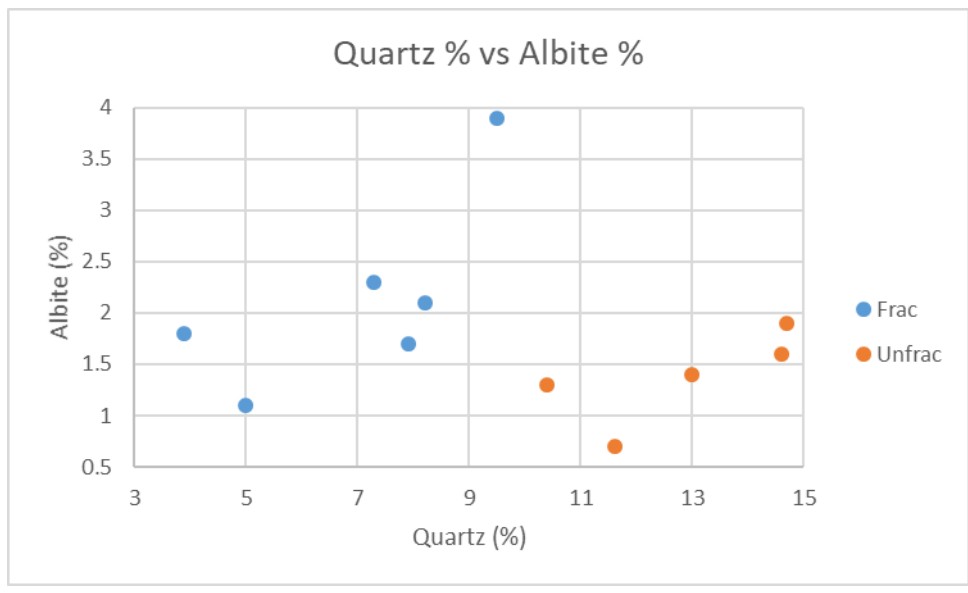


**Figure 7: Quartz content versus albite content.**





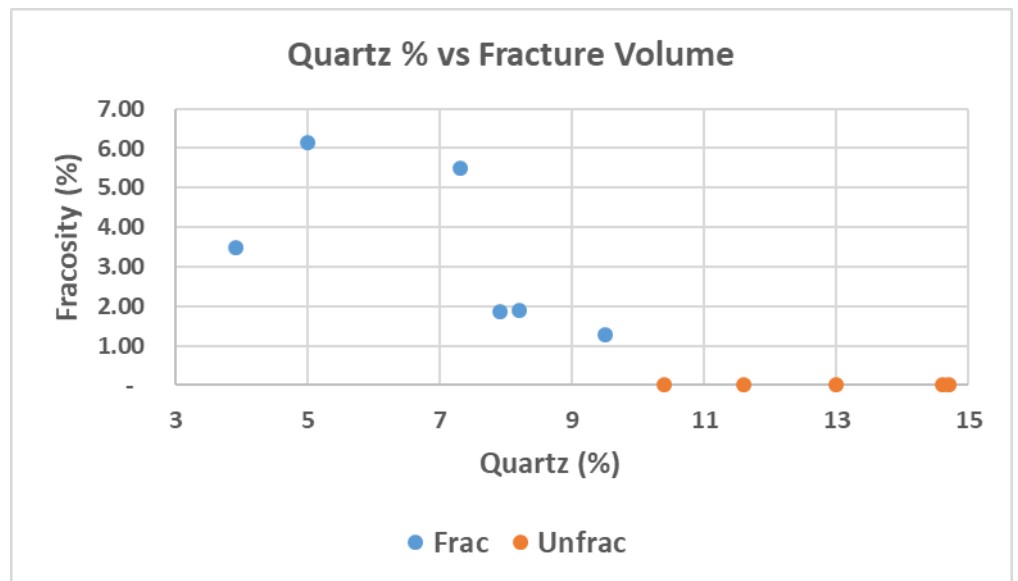

**Figure 8: Fracture volume and quartz mineral percent of fractured and unfractured layers.**

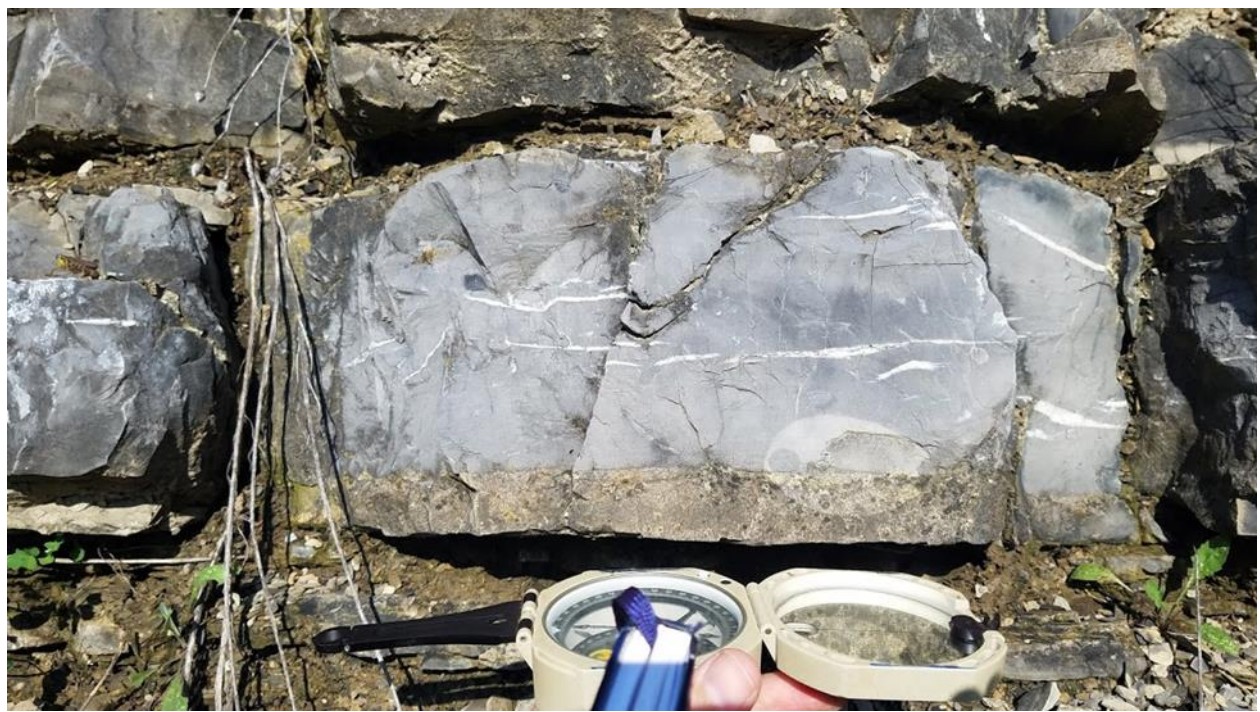


**Figure 9: Field photo of fractured layer 2.**



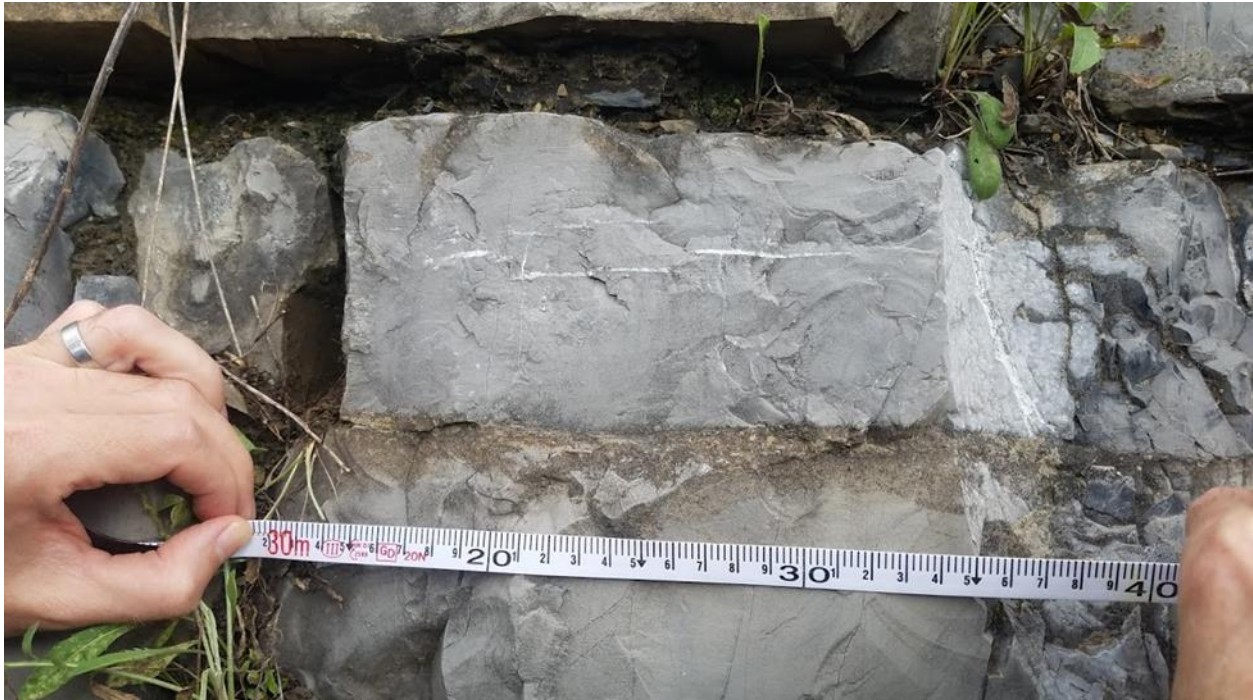

**Figure 10: Field photo of fractured layer 6.**





**Figure 11: Petrographic photograph of fractured layer 1 showing low birefringence silica grain.**





Figure 12: Petrographic photograph of the silicified, unfractured base of fractured 2.







Figure 13: Petrographic photograph of fracture layer 1 possible radiolarian.





**Figure 14: Petrographic photograph of the layer boundary for the base of layer 2(left side) and fractured layer 2.**





**Figure 15: Petrographic photo of fractured layer 1 fracture with cross-cutting relationships. The wide, horizontal fracture in this field of view is bed-parallel.**



## Appendix 1:

### XRD Mineral Composition

| Layer | Calcite | Quartz | Albite | Pyrite | Muscovite | Kaolinite | Dolomite | Vermiculite | Fracosity % |
|-------|---------|--------|--------|--------|-----------|-----------|----------|-------------|-------------|
| F1 | 93.8 | 3.9 | 1.8 | 0.5 | | | | | 3.49 |
| U1.5 | 84.1 | 10.4 | 1.3 | | 1.5 | 2.7 | | | |
| 2-base | 80.4 | 15 | 0.7 | | 3.9 | | | | |
| F2 | 93.7 | 5 | 1.1 | 0.1 | | | | | 6.15 |
| U2.5 | 82.9 | 13 | 1.4 | | 2.2 | 0.6 | | | |
| CA2 | 62.1 | 11.4 | 0 | | 24.7 | | | 1.7 | |
| F3 | 90.2 | 7.3 | 2.3 | 0.3 | | | | | 5.48 |
| U3.5 | 83.5 | 14.7 | 1.9 | | | | | | |
| F4 | 86.3 | 9.5 | 3.9 | 0.3 | | | | | 1.29 |
| U4.5 | 87.7 | 11.6 | 0.7 | | | | | | |
| F5 | 89.7 | 8.2 | 2.1 | 0.1 | | | | | 1.88 |
| U5.5 | 83.8 | 14.6 | 1.6 | | | | | | |
| CA6 | 61.5 | 17.8 | 1.7 | | 18.7 | | | 0.3 | |
| CA6 decarb | 0 | 0 | 0 | | | | | | |
| 6-base | 72.8 | 17.1 | 1.3 | 1.7 | | | 7 | | |
| F6 | 89.7 | 7.9 | 1.7 | 0.6 | | | | | 1.85 |
| 6-top | 84.4 | 14.4 | 1.2 | | | | | | |

## Appendix 2:

### Volume Loss Calculation

Volume loss calculations were used to analyze whether the volume that migrated from fractured layers could explain the fracosity amount observed. The volume represented by fractures within any fractured layer was hypothesized to be an estimate of the amount of silica that migrated from the fractured layers to the silicified base layers. We began by assuming that the distribution of silica within each bed was initially homogeneous. For fractured layer 6, fractured layer 2, 6 base layer, and 2 base layer, we then assumed that the final distribution of silica within the fractured layers (F2 and F6) and their underlying base layers (F2-base and F6-base) resulted from silica migration during the transition from opal to quartz. The mineral weight percent generated by XRD was then converted to volume percent using the following relation:

$$volume \, \% \, quartz = weight \, \% \, quartz \, \times \, \frac{\rho \, rock}{\rho \, quartz}$$

Where $\rho$ denotes density. The density of the rock was calculated as the reciprocal of:

$$\frac{quartz \, wt \, \%}{\rho \, quartz} + \frac{albite \, wt \, \%}{\rho \, albite} + \frac{pyrite \, wt \, \%}{\rho \, pyrite} + \frac{dolomite \, wt \, \%}{\rho \, dolomite} + \frac{calcite \, wt \, \%}{\rho \, calcite}$$





By using the density of the rock, we can then estimate the volume percent of quartz assuming it was initially homogeneous. The initial volume percent of quartz ($V_q$) is calculated using the thicknesses of the fractured layers ($T_f$) and the silicified base layers ($T_b$):

420

$$V_q = vol\ \%\ in\ fractured\ layer\ \times \frac{T_f}{T_f + T_b} + vol\ \%\ in\ base\ layer\ \times \frac{T_b}{T_f + T_b}$$

The volume of quartz currently hosted in the fractured layer was subtracted from $V_q$. This difference is the percent of volume loss assuming the content in the fractured layer was quartz; this is then divided by 0.79 assuming the migrated material was dissolved biogenic silica (Hooker et al., 2017). This calculation produces a total volume loss of biogenic silica transitioning to quartz. An example calculation from layer 6 is as follows:

425

$$\text{Quartz} \quad \text{albite} \quad \text{pyrite} \quad \text{dolomite} \quad \text{calcite}$$

$$\frac{0.171}{2.65} + \frac{0.013}{2.62} + \frac{0.017}{5.01} + \frac{0.007}{2.84} + \frac{0.728}{2.71} = 0.366$$

$$\frac{1}{0.366} = 2.73\ rock\ density\ in\ g/cc$$

$$0.171 \times \frac{2.73}{2.65} = 0.176\ volume\ \%\ quartz$$

$$(0.176 \times 0.5) + (0.0808\ \times 0.5) = 0.1284 = 12.84\ \%\ V_q$$

430

$$12.84 - 8.08 = 4.76\ \%\ quartz\ volume\ loss$$

$$\frac{4.76}{0.79} = 6.02\ \%\ opal\ volume\ loss$$