# Peer review of "Silica diagenesis-driven fracturing in limestone: an example from the Ordovician of Central Pennsylvania"

_Solid Earth, 2020_

## Referee Comment (RC1) · Anonymous Referee #1 · 20 May 2020

Emily M. Hoyt and John N. Hooker

**Anonymous Referee #1**

The authors provide observations on bed parallel calcite veins and link their formation
to silica diagenesis. Geochemical fracturing is not well studied and of interest to read-
ers and fits the scope of SE. This work builds on previous work by the second author,
the orientation of fractures and the geologic setting is different. As presented the con-
clusions are not supported and are hypothesis-driven. Abstract lacks specific informa-
tion and uses broad generalizations to describe the outcrop and fractures. The abstract
needs an opening statement(s) that help the reader identify the topic/hypothesis/or
problem being evaluated in the paper – this could be achieved with the addition of
specific details on lithologies, fracture type, and distribution. Methods section requires
additional information Point quantification – explains the use of a grid to count fractures

and optical petrography was done but no explanation of what was done besides using a microscope – did point counting occur? For example: Line 199-200 claims petrographic evidence of show significant amount of detrital quartz, the slide shows 1 grain, no point count data given. XRD cannot distinguish between detrital and biogenic. This is an unsupported claim. The difference in mechanical properties between layers is dismissed because fractures are present in brittle limestone layers and absent from intervening shales — a literature review would suggest this is in fact due to mechanical difference between the shale and limestone Calculations presented are based on generalized assumptions of rock mechanical properties The manuscript lacks citations throughout – well-documented procedures exist for quantifying fractures no need to Authors claim to calculate fracture volume – a volume requires 3D all counts are done in 2D and are listed as area's elsewhere in the document The word "fracosity" is not required and authors should present background information and data that supports the creation and use of a new term and/or the reason for not using a well-established method for documenting fracture distributions within a rock mass and associated terminology. The manuscript lacks introduction to stratigraphy and, The tectonic events are eventually mentioned but both stratigraphy and geologic history/tectonic setting needs to be more detailed and happen earlier in the manuscript. A larger regional map would be helpful inset to the geologic map. The manuscript is hard to follow partway through the author's voice and word choice changes, perhaps it was written in two separate parts and joined with the latter part being more polished (from ∼line 250 writing style changes). Manuscript mixes results and observations with interpretation throughout

The language used throughout the early portion of the manuscript is not precise (until ∼line 250). • Manuscript lacks data/quantification that supports the use of terms like "a highly stratified fracture pattern" • Distracting editorial typos exist throughout • Discussion refers to observations and a process never mentioned in results • Introduction of methods for fracture formation should move earlier in the manuscript, they first appear in the discussion Stratigraphy is defined as Limestone, Shale, Argillaceous Limestone, Clay, and fractured layer – this is over generalized and a mix of lithology,

grain size, and deformation features. The authors should provide an explanation of lithologies and how they were determined. What type of limestone and how do limestone and argillaceous limestone differ; how does clay (a grain size or mineral group not a rock type) differ from shale -what type of types of cement are present — does the fracture layer also have a lithology? Do fractures refer to all fractures or just the calcite veins if the study is on the calcite veins refer to them as calcite veins Figure captions are lacking and do not allow the figure to stand alone; photomicrographs are poor quality and unlabeled. Table 1 – inconsistent presentation of data Fractured layers have no detail regarding rock No – paper lacks citation throughout especially in the geologic setting and methods of determining fracture distribution

---

## Referee Comment (RC2) · Vincenzo Guerriero (Referee) · 19 Jun 2020

GENERAL COMMENTS ABOUT THIS STUDY In the submitted paper authors provide an interesting field example of bed parallel joint sets in limestone and furnish an attractive explanation of a possible jointing model which is based on chemically driven fracturing (namely, silica diagenesis). The study is well reasoned and informative, as well as corroborated by field data. As this topic may be of interest for a wide audience of potential readers, I would advise publication of this manuscript after a minor revision, according to the following comments. SPECIFIC COMMENTS - It is unclear to me why authors state as tectonic strain or fluid overpressure, hardly can explain horizontal joint formation. Although a clear negative correlation between silica content and occurrence of horizontal joints is evident, I wonder whether this is sufficient in order to exclude the effects of a compressive tectonic and fluid overpressure or a combined effect of these latter and chemical driven fracturing. Authors should further clarify this point. - Introduction: although Introduction is rather illustrative, authors should depict more thoroughly here the state of the art in the field of chemically driven fracturing. - Section 'Point Quantification': the described method of fracture porosity quantification is interesting, nevertheless authors may illustrate a comparison with more traditional methods such as scan line and scan area and the (eventual) advantages of the use of this criterion. - Section 'Point Quantification': It's not clear to me whether the described method is applied over field or thin section images. Furthermore, authors should clarify if a lower threshold is adopted for fracture size (according to Ortega et al., 2006; Guerriero et al., 2010) or all existing fractures within the sampled area are accounted . - Section 'Point Quantification': authors may explain more thoroughly this method. Maybe, an illustrating picture might be opportune here. - Section 'Point Quantification': it's not clear to me why authors prefer the use of fracosity, which provides an estimation of porosity associated to all fractures falling within the investigated image/area, rather than fracture porosity associated to a single fracture set (e.g. horizontal set). - Sections 'Introduction', 'Discussion' and 'Conclusions': such sections are well reasoned and illustrative, nevertheless I suggest to point out more explicitly as horizontal and vertical fracturing are two mechanical problems substantially different. In orthogonal to bedding jointing, layers of different size and mechanical properties are constrained to show the same horizontal strain whereas, in case of horizontal jointing, the involved beds are independent mechanical systems. As a consequence, all theories about joint filling are here unusable. - Section 'Discussion': authors provide an interesting strain energy based analysis in order to justify a vertical extension of horizontally fractured beds, nevertheless it is unclear to me how the two cited terms Sf and Sc are compatible. In my opinion we can observe two alternative scenarios: (i) Sc different form 0 and Sf = 0, in case of contraction strain and (ii) Sf different form 0 only in case of extensional

strain. Probably I have misunderstood something in discussion and, so, I suggest to make clearer such section. - Section 'Discussion': It would be interesting to consider the following as a possible alternative model for horizontal jointing: under hypothesis of early silica migration, and consequent shrinking, some internal/residual stress can be induced by heterogeneous chemical induced alterations. Namely, if some portion of a rock layer experiences shrinkage, the remaining part may bear the overburden, so prohibiting vertical contraction. This may justify the condition of vertically pinned rock layer. Should be noted that, as frequently rock compressive strength is larger than tensional strength of about one order of magnitude, a small portion of a rock layer (slightly over the 10% of the total) which is not subjected to shrinkage is sufficient to induce tensional fracturing within the remaining part of such layer.

MINOR TECHNICAL CORRECTIONS - line 62 – 63: the word 'layers' appears too much times within one sentence. - line 173: 'joint abundance' is rather generic if this term denotes a function of bed thickness; the sentence 'joint spacing is proportional to bed thickness' is more appropriated; - line 181: 'hydrostatic pressure' is generic, whereas 'overburden stress' is more opportune; authors may substitute with: '…a state where the fluid pressure exceeds the overburden stress at a given depth. In this latter instance horizontal fractures can form …'; - line 273: the sentence '… the host rock is prohibited from vertical contraction …' may be misleading as it might lead the reader to think about some boundary conditions or external constraint which hinder subsidence or vertical compaction; - Fig. 14: It is not clear to me whether or not the observed fractures are orthogonal to bedding. - Table 2: also in table captions authors might explain that Total Area is expressed as number of all grid intersections falling within the inspected image.

CITED REFERENCES Ortega, O., Marrett, R., Laubach, E., 2006, Scale-independent approach to fracture intensity and average spacing measurement. AAPG Bulletin 90, 193–208. Guerriero, V., Iannace, A., Mazzoli, S., Parente, M., Vitale, S., Giorgioni, M., 2010. Quantifying uncertainties in multi-scale studies of fractured reservoir analogues:

implemented statistical analysis of scan line data. Journal of Structural Geology 32, 1271-1278, doi:10.1016/j.jsg.2009.04.016

With my regards and best wishes for your work, Vincenzo Guerriero

---

## Referee Comment (RC3) · Vincenzo Guerriero (Referee) · 19 Jun 2020

The described method of fracture porosity quantification is interesting, nevertheless authors may illustrate a comparison with more traditional methods such as scan line and scan area and the (eventual) advantages of the use of this criterion.

It's not clear to me whether the described method is applied over field or thin section images. Furthermore, authors should clarify if a lower threshold is adopted for fracture size (according to Ortega et al., 2006; Guerriero et al., 2010) or all existing fractures within the sampled area are accounted.

[Figure]

Furthermore, it is not clear to me why authors prefer the use of fracosity, which provides an estimation of porosity associated to all fractures falling within the investigated image/area, rather than fracture porosity associated to a single fracture set (e.g. horizontal set), in order to analyze a possible correlation between fracture intensity and silica content data. Best regards, Vincenzo Guerriero

---

## Referee Comment (RC4) · Vincenzo Guerriero (Referee) · 19 Jun 2020

It would be interesting to consider the following as a possible alternative model for horizontal jointing: under hypothesis of early silica migration, and consequent shrinking, some internal/residual stress can be induced by heterogeneous chemical induced alterations. Namely, if some portion of a rock layer experiences shrinkage, the remaining part may bear the overburden, so prohibiting vertical contraction. This may justify the condition of vertically pinned rock layer. Should be noted that, as frequently rock compressive strength is larger than tensional strength of about one order of magnitude, a small portion of a rock layer (slightly over the 10% of the total) which is not subjected to

shrinkage is sufficient to induce tensional fracturing within the remaining part of such layer. Best regards, Vincenzo Guerriero
* * *

---

## Referee Comment (RC5) · Vincenzo Guerriero (Referee) · 20 Jun 2020

I am slightly perplexed about use of a neologism (i.e. fracosity) which seems unnecessary here, as it simply denotes a fracture porosity estimation. Furthermore, area based estimations of porosity need to be carefully dealt with; in case of primary porosity (i.e. pore related, under hypothesis of homogeneous isotropic pore distribution) area and volume porosity assumes the same value, i.e. Pore_Area/Total_Area = Pore_ Volume /Total_Volume, whereas, in case of strong anisotropic void distributions (i.e. also fracture porosity), some correction is needed. This latter is based on reciprocal orientations of scan line or area and analyzed fracture set. In this manuscript such a correction was

not applied or explained.

---

## Short Comment (SC1) · 22 Jun 2020

Dear Referee,

We thank you for your time that you invested in reading our paper and submitting such productive criticism. We have reviewed your comments and will use them to enhance our paper's presentation and content. We are currently visiting the structure, grammar, and background issues you mention and improving them to ensure a streamlined and detailed presentation that can be well-understood by all readers. In the spirit of open discussion, we would also like to comment on a few of your remarks as follows:

[Figure]

1. Conclusions not supported/ hypothesis-driven: Our approach was to present multiple hypotheses that could potentially explain the observed fractures. Although we cannot concretely prove that any one of our hypotheses is the mechanism for our stratified fracture set, most proposed hypotheses fail our tests, such that we are left with the concept of silica diagenesis. We cannot disprove this explanation, and as well, it is supported by our petrographic and XRD data results. To summarize, we allow that we cannot prove silica diagenesis was the cause, but our main aim was to disprove other hypotheses and suggest the viability of silica diagenesis as an alternative. We welcome arguments that either (i) uphold the viability of alternative hypotheses, or (ii) negate the viability of silica diagenesis.

2. Optical petrography/ point counting: We would like to clarify, and will also revise this in the presentation of our paper, that we performed point-counts on field photos and not on thin sections; but we otherwise used the standard method that is normally applied to thin section images. We would also like to explain that our quartz composition percentages were quantitatively derived using XRD analysis. We agree with the referee that XRD cannot distinguish diagenetic from detrital quartz; let us clarify that our interpretations of the type of quartz (biogenic or detrital) was based on qualitative petrographic observations.

3. Terminology: fracosity: We recognize the potential misunderstanding and unfamiliarity of this new term. We would like to clarify that the use of this term is to emphasize the difference between fracture porosity, which is not our focus in this paper, and our method of point-counting field photographs. Fracosity differs from the conventional method of scanlines used to calculate fracture intensity in that fracosity represents the volume fraction of the host rock that is occupied by fractures, including fracture cement. This quantity was determined by dividing the fracture points in the field photo by the sum of the fractured and unfractured points in the photo and then multiplying that by 100 to produce a percent. In that sense it is an area measurement, just as porosity, although a volumetric fraction by definition, is commonly measured as an area in point

counts of thin sections. We concede that without spending more time on the subject, it is perhaps not wise to go with a little used term, but on the other hand, we are unsure of another term that encapsulates the volume fraction of rock comprising fracture pore space and cement. We will be sure to focus on this issue to improve how we introduce and reference the term, or abandon it, to avoid such confusion. We welcome opinions on the matter.

Again we thank you for your thoughtful review, which will clearly improve the work overall.

Sincerely,

Emily Hoyt and John Hooker

---

## Author Comment (AC1) · 3 Jul 2020

Dear Dr. Guerriero,

Thank you for your interest in our paper and providing us with such constructive criticism. We have reviewed your remarks and will implement them to substantially improve our paper. We are currently visiting the structure, grammar, and background information issues that you mention and will be applying modifications to guarantee a cohesive and thorough presentation. We would also like to briefly address your primary concerns as follows:

[Figure]

1. Discounting a mechanism of tectonic strain and fluid overpressure

a. Clearly, we cannot categorically exclude tectonic strains and fluid pressure as important factors in the development of the fractures we are considering. However, elsewhere in the outcrop we find the classic limestone bedding-bound arrays of fractures, which are absent in intervening shales. Such a pattern is an excellent example of a set of fractures generated in response to tectonic stretching. Critically, these fractures are present within multiple stacked limestone beds, and again, are absent in intervening shales. In contrast, the horizontal fractures are present within individual limestone beds, over distances of many tens to hundreds of meters, and completely absent in over- and under-lying limestone beds. Granted, the horizontal fractures have a different orientation, but in the end, if they are driven by tectonic strains (assisted by fluid overpressures or not) we find it difficult to explain why these fractures should be so conspicuously bound to certain limestone layers. The consistently low silica contents in these fractured beds points us to a chemical mechanism for the fractures. See also response #3 below.

2. Terminology: fracosity

a. We understand the confusion that has arisen with the introduction of this term. Therefore, we would like to clarify that the use of this term is to emphasize the difference between fracture porosity, which is not our focus in this paper, and the total volume fraction of rock comprising fracture porosity and cement, which is. Our measurement is dimensionally identical to the P22 of Dershowitz and Herda (ARMA, 1992) in that it is the dimensionless ratio of fracture-area to unit rock area. This measure was called "fracture porosity" in that study because it is analogous to porosity measured in 2D thin sections, and indeed our measurement would be exactly the same as fracture porosity except that it is not porosity, because these fractures are essentially entirely filled by cement.

b. Part of the confusion stems from our scant description of our method of point-
counting field photographs. We will be improving our description of our method used to quantify the fracturing using point quantification—that is, gridded point counts overlain on field photographs–to better describe our method.

c. Fracosity does differ from the conventional method of scanlines used to calculate fracture intensity, in that fracosity is a point-count-based areal measurement of the host rock that is occupied by fractures. You remind us that this measurement, like scanlines, is not necessarily equal to the true P33 (volumetric fracture porosity, including fracture cement in this case), because the observation surface may lie at some low angle to fractures. Indeed we were remiss to neglect that point, and will clarify in our upcoming revision.

d. We therefore concede that our methods were too hastily described to adequately introduce a new term, and will decide whether to abandon the new term or to more thoroughly introduce it for the final draft.

3. Horizontal and vertical jointing

a. You bring up a good point, that joints lying parallel to their host beds can be thought of as an independent system of fractures, whereas bedding-orthogonal joints can be more reasonably expected to show similar systematics across multiple beds. Indeed, that is exactly what we see in the field, and one could argue that the presence of joints in one bed simply reflects that bed's physical characteristics, or even just the vagaries of tectonic strain. However, we again note the striking inconsistency of fracture abundance among beds having only subtle mineralogic variation (i.e., they are all limestones). Moreover, the fractures in question are quite short (i.e., they have small tip-to-tip distances). If the beds in question hosted a small number of very long fractures, then we could posit that chance alone resulted in the observed variation in fracture abundance. But the large number of fractures, many of them orthogonal to bedding, stongly suggests that something about those beds has primed them to undergo fracturing, and of course the XRD evidence suggests that that something is silica

diagenesis.

4. Strain energy analysis

a. We would like to clarify that we postulated an energy-balance scenario for different cases of vertical contraction that can be considered. You question whether SF and SC are compatible within the same deforming rock system. Indeed, principally the energy comparison is likely between SC versus G and H—the energetic cost of layer collapse versus the cost of fracturing (G) and the benefit of collapse (H). We simply wanted to mention SF for completeness, as the opening of veins entails a shape change and thus some finite energetic cost. We agree that this cost is likely negligible compared to the other terms.

b. Furthermore, in a followup comment, you suggest a more specific mechanism of vertical pinning, whereby some portion of the rock mass is prevented from collapse, producing fractures within the remaining rock mass. We find this idea to be compatible with our energy balance approach: the prohibition from vertical contraction is, in essence, a large energetic cost to vertical contraction, SC. This is a good way to describe a potential pathway for the development of the fractures, and so we will incorporate this comment into our revision.

Again, we are most grateful to Dr. Guerriero, and Reviewer #1, for your hard work and excellent suggestions. We hope that these quick responses are satisfactory for now, as we work toward incorporating the comments into our final manuscript.
* * *

---

## Author Comment (AC2) · 3 Jul 2020

My apologies for the confusion, but the previous Short Comment was submitted by the Authors, but inadvertently using the wrong ID. Please see that comment for our initial response to Reviewer 1. Again we thank the reviewer for his or her careful reading of our work, and helpful suggestions for improvement, and we will be preparing a final response along with our revised manuscript, in the coming weeks.

Sincerely,

John Hooker and Emily Hoyt